

**Sequential Data Assimilation for Real-Time Probabilistic**
**Flood Inundation Mapping**
Keighobad Jafarzadegan[*], Peyman Abbaszadeh, Hamid Moradkhani
Center for Complex Hydrosystems Research, Department of Civil, Construction, and
Environmental Engineering, University of Alabama, Tuscaloosa, AL
*Corresponding author: kjafarzadegan@ua.edu





## Abstract

Real-time probabilistic flood inundation mapping is crucial for flood risk warning and decision making during the emergency of an upcoming flood event. Considering high uncertainties involved in the modeling of a nonlinear and complex flood event, providing a deterministic flood inundation map can be erroneous and misleading for reliable and timely decision making. The conventional flood hazard maps provided for different return periods cannot also represent the actual dynamics of flooding rivers. Therefore, a real-time modeling framework that forecasts the inundation areas before the onset of an upcoming flood is of paramount importance. Sequential Data Assimilation (DA) techniques are well-known for real-time operation of physical models while accounting for existing uncertainties. In this study, we present a Data Assimilation (DA)-hydrodynamic modeling framework where multiple gauge observations are integrated into the LISFLOOD-FP model to improve its performance. This study utilizes the Ensemble Kalman Filter (EnKF) in a multivariate fashion for dual estimation of model state variables and parameters where the correlations among point source observations are taken into account. First, a synthetic experiment is designed to assess the performance of the proposed approach, then the method is used to simulate the Hurricane Harvey flood in 2017. Our results indicate that the multivariate assimilation of point-source observations into hydrodynamic models can improve the accuracy and reliability of probabilistic flood inundation mapping by 5-7% while it also provides the basis for sequential updating and real-time flood inundation mapping.

**Keywords:** Data Assimilation; Probabilistic Flood Inundation Mapping; Hydrodynamic Model; Ensemble Kalman Filter



## 1. Introduction


The on-time, accurate, and reliable characterization of an upcoming flood event is imperative for
proper decision making and risk analysis. A well-calibrated hydrologic model coupled with
reliable weather forecast models can be used to generate the streamflow forecast (Clark and Hay,
2004; Cuo et al., 2011; Habets et al., 2004). While streamflow forecasting during flood events is
indispensable, the critical step for flood risk analysis is to estimate the flood inundation areas
corresponding to the forecasted streamflow of a potential upcoming event. Hydrodynamic models
are common tools used to simulate the physics of a river system and predict the spatiotemporal
distribution of water surface elevation. The predicted water surface elevation can be simply
converted to water depth and inundation area by overlaying with a high-resolution Digital
Elevation Model (DEM) (Merwade et al., 2008; Teng et al., 2017).
According to the literature, most studies have analyzed the flood events for which the flood extent
maps were available from surveying or satellite remote sensing. These studies include but are not
limited to, calibration and assimilation of hydrodynamic models (Baldassarre et al., 2009; García-
Pintado et al., 2013; Gobeyn et al., 2017; Hostache et al., 2009; Lai et al., 2014; Pappenberger et
al., 2007; Rahman and Thakur, 2018; Tarpanelli et al., 2013). Depending on the research
objectives, such studies are crucial as they address important theoretical questions and advance the
flood modeling task. For example, several studies have used satellite remote sensing data, such as
Synthetic Aperture Radar (SAR) images, to find the sensitivity of hydrodynamic models to their
parameters, compare calibration strategies and test the application of assimilating remote sensing
data into these models (Di Baldassarre et al., 2009; Hunter et al., 2005; Mason et al., 2009; Matgen
et al., 2010). Since floods happen in a short period and at a certain location, it is most often not
possible to find an appropriate remote sensing image that covers those inundated areas during the





flood period. This is the main reason that research on flood inundation mapping is mostly limited
to post-event analysis where specific study areas with available remote sensing data are used as
testbeds.
Federal Emergency Management Agency (FEMA) is the leading agency in the United States that
provides flood hazard and risk maps over the Contiguous United States (CONUS). These maps
display the flood-prone areas corresponding to specific return periods (e.g. 100 and 500-year
events). While the FEMA flood hazard and risk maps provide general information about risk areas,
they are not always reliable for an upcoming flood event with different return periods. For
example, FEMA 100-year and 500-year flood hazard maps covered only one-third and half of the
inundated areas induced by Hurricane Harvey in Harris County, Texas, respectively (Pinter et al.,
2017). The National Water Center Innovators Program proposed the idea of real-time flood
inundation mapping across the United States in 2015 (Maidment, 2017). It highlighted the
importance of event-based flood inundation mapping where a model uses the forecasted river
discharge to estimate the inundation areas corresponding to a specific flood just before the onset
of the event.  Compared to the traditional flood hazard mapping, real-time flood inundation
mapping is more informative and beneficial for emergency response-related decision-making.
In real-time flood inundation mapping, the model takes advantage of forecasted forcing data and
generates inundation areas corresponding to an upcoming flood event. Providing these maps ahead
of time is extremely valuable for building a robust flood warning system. Data assimilation (DA)
is an effective approach commonly used to improve the performance of real-time hydrologic
forecasting by updating the model state variables and parameters when new observation becomes
available (Moradkhani et al., 2019). The integration of DA with physical models is highly
advantageous as it enables accounting for different sources of uncertainties involved in model


predictions. These include  (1) forcing data uncertainty due to the limitation of measurements and
spatiotemporal representativeness of the data (Alemohammad et al., 2015; Kumar et al., 2017), (2)
parameter uncertainty due to equifinality and non-uniqueness of parameters (Abbaszadeh et al.,
2018; Leach et al., 2018), (3) model structural uncertainty due to the imperfect representation and
conceptualization of a real system (Abbaszadeh et al., 2019; Pathiraja et al., 2018; Zhang et al.,
2019) and (4) initial and boundary condition uncertainty (DeChant and Moradkhani, 2014; Lee et
al., 2011).
Probabilistic forecasting and uncertainty quantification using DA have been the core of modeling
in the atmospheric and oceanic sciences (e.g. Anderson and Anderson, 1999; Courtier et al., 1993).
Later, the hydrologic community started to utilize this approach to account for the uncertainties
involved in different layers of model predictions and provide more accurate and reliable model
estimates such as soil moisture (Pauwels et al., 2001; Reichle et al., 2002), streamflow
(Moradkhani et al., 2005a; Vrugt et al., 2006), snow (Sheffield et al., 2003; Slater and Clark, 2006)
and so many other variables. Despite these advances in hydrologic studies, the application of data
assimilation in conjunction with hydrodynamic models has received little attention in the literature.
The characterization of uncertainty in hydrodynamic models for probabilistic flood inundation
mapping has been mostly limited to conventional techniques, such as random Monte Carlo
sampling (Domeneghetti et al., 2013; Neal et al., 2013; Pedrozo-Acuña et al., 2015; Purvis et al.,
2008) and Generalized Likelihood Uncertainty Estimation (GLUE) (Aronica et al., 2002a;
Romanowicz and Beven, 2003).
The effectiveness and application of assimilating remotely sensed data (e.g. Soil Moisture Active
Passive (SMAP)) into hydrologic models have been vastly investigated in the literature
(Abbaszadeh et al., 2020; Azimi et al., 2020; Lievens et al., 2017). However, given the small scale





of the hydrodynamic modeling process, the spatiotemporal resolution of current satellite products
is not adequate for assimilating into these models. To properly estimate the flood inundation
extent, a spatial resolution less than river width (e.g. 100 m) is recommended. In addition, due to
the short duration of floods, satellite data with daily revisit time is needed. Since remote sensing
products do not provide such high spatiotemporal resolution data for hydrodynamic models, the
research on hydrodynamic data assimilation is limited in the literature. Due to the coarse spatial
resolution of satellites that provide water surface elevation data, some studies have limited their
analyses to large rivers with a width of above 1 km (e.g. study of Nile and Amazon) (Brêda et al.,
2019). However, since the width of the majority of rivers is less than 100 meters, these studies
cannot be practically used in many regions. Several studies used higher resolution synthetic
SWOT data to evaluate the performance of assimilation techniques (Durand et al., 2008; Munier
et al., 2015; Pedinotti et al., 2014; Yoon et al., 2012). While these works provided important
information about the assimilation of satellite data into hydrodynamic models, their applications
are only limited to synthetic experiments, making them impractical for real case studies. Some
studies have implemented indirect methods to estimate WSE from flood extents generated by high-
resolution SAR satellite data (Giustarini et al., 2011; Hostache et al., 2010; Matgen et al., 2010b;
Neal et al., 2009). This approach can provide high-resolution data that is suitable for the majority
of rivers. However, the reliability of this data is concerning because the methods used to convert
the flood extent to WSE pose additional errors which downgrades the quality of the final observed
data for assimilation practices. Besides these issues, the major drawback of remote sensing data
assimilation pertains to their coarse temporal resolutions. To efficiently monitor the flood
dynamics, the assimilation process should be performed at a daily/hourly time scale, however, the
revisit frequency of satellites used for capturing the water surface elevation ranges from a week to





a month. Therefore, there is a significantly low chance to capture multiple real-time remote sensing
images for the majority of inundated catchments during flood events. In the most optimistic
scenario, assimilation of satellite data is only limited to one/two updates during the simulation
period which may not be sufficient for reliable probabilistic flood inundation mapping.
Application of DA in hydrodynamic modeling can be either river monitoring or flood inundation
mapping. The goal of hydrodynamic data assimilation for river monitoring is to track variations in
the channel roughness and bathymetry in the long run. Therefore, the weekly/monthly satellite
data can be well assimilated into the models as the channel characteristics do not change on a daily
basis. On the other hand, flood inundation mapping needs an hourly/daily track of WSE because
floods happen rapidly and affect the river dynamics on a short time scale. The literature indicates
those studies that assimilated data into hydrodynamic models have been mostly designed for river
monitoring (Brêda et al., 2019; Durand et al., 2008; Yoon et al., 2012b). To capture the daily
dynamics of the rivers for real-time flood inundation mapping, the discharge and water stage
values measured at the gauge stations can be assimilated into the hydrodynamic models. Xu et al.,
(2017) performed a Particle Filtering (PF) approach to assimilate the water stage data from six
gauges into a hydrodynamic model. In order to calculate the particle weights in the filtering
process, they assumed that gauge observations are independent. In this study, however, we
consider interconnections among the gauge stations and apply multivariate Ensemble Kalman
Filter (EnKF) to a 2D hydrodynamic model for better characterization and quantification of
uncertainty and further improving the accuracy of model simulations.
Advancing the probabilistic hydrodynamic modeling with DA techniques is a necessary step to fill
the gap between hydrology and hydrodynamics. To address this problem, this study aims to
explore the capability of a standard sequential DA technique, namely the EnKF, for real-time





probabilistic flood inundation mapping. The past studies that used DA in conjunction with
hydrodynamic models have mostly focused on the quantification of uncertainty in one or two
hydrodynamic variables (e.g. Giustarini et al., (2011) and Hostache et al., (2018) only investigated
the uncertainty in the upstream flow and rainfall respectively; Yoon et al., (2012) focused on the
uncertainty of river bathymetry while ignoring the roughness parameter uncertainty). In addition,
the main application of DA-hydrodynamic modeling framework has been in river monitoring at
long-term or water stage forecasting during the flood events (Brêda et al., 2019; Matgen et al.,
2010; Xu et al., 2017). However, this study takes one step further and proposes a DA-
hydrodynamic modeling framework for real-time probabilistic flood inundation mapping while
accounting for all sources of uncertainties involved in the model simulations. These include
hydrodynamic model parameters (channel roughness and river bathymetry) uncertainty, forcing
data (river boundary conditions) uncertainty, and state variable (water depth) uncertainty.
Additionally, unlike past works that assimilated either discharge or water stage into the
hydrodynamic model, this study performs a multivariate DA to incorporate the observed values of
both variables into the hydrodynamic model for a reliable simulation of flooding and its
corresponding inundation area.

## 2 Data and Study area

In this study, we simulate the Hurricane Harvey flood, one of the worst natural disasters in the
history of the United States that caused more than 120 billion in damage
(https://www.nhc.noaa.gov/data/tcr/AL092017_Harvey.pdf). The Harvey storm hit Texas on
August 25, 2017, caused massive precipitation for six continuous days and resulted in extreme
flooding condition in Houston and surrounding areas. Given the considerable uncertainties in
hydrologic and hydrodynamic processes of such an extreme flood, a deterministic modeling





approach with fixed inputs provides erroneous simulations that are highly different from
observations. To account for the uncertainties involved in different layers of flood simulation, this
study implements a DA-hydrodynamic modeling framework and provides probabilistic flood
inundation maps.
Figure 1.a shows the study area that consists of four main channels (blue lines) and eight tributaries
(red lines). The upstream and downstream boundary conditions (purple points) are provided from
daily streamflow in four USGS gauges ((#08068090, # 08068500, #08068740, #08068780) and
water stage time series at the downstream gauge (#08069500). The daily streamflow discharge in
two internal gauges (green points #08068800 and #08069000) and water stage time series in the
second internal gauge are the observations that will be assimilated into the LISFLOOD-FP model.
Figures 1.b and 1.c present the geographic location of the study area within the state of Texas and
San Jacinto watershed, respectively. To set up the LISFLOOD-FP model, we use a DEM with 120
m spatial resolution. Such a coarse resolution DEM alleviates the computational intensity of the
proposed probabilistic hydrodynamic modeling framework. It should be noted that the subgrid
solver used for simulation of flood has the advantage of accepting narrow rivers with a width of
less than 120 m while the cell sizes are 120 m. In this study, the DA-hydrodynamic modeling
framework is parallelized and performed on the University of Alabama High-Performance
Computing (UAHPC) cluster.

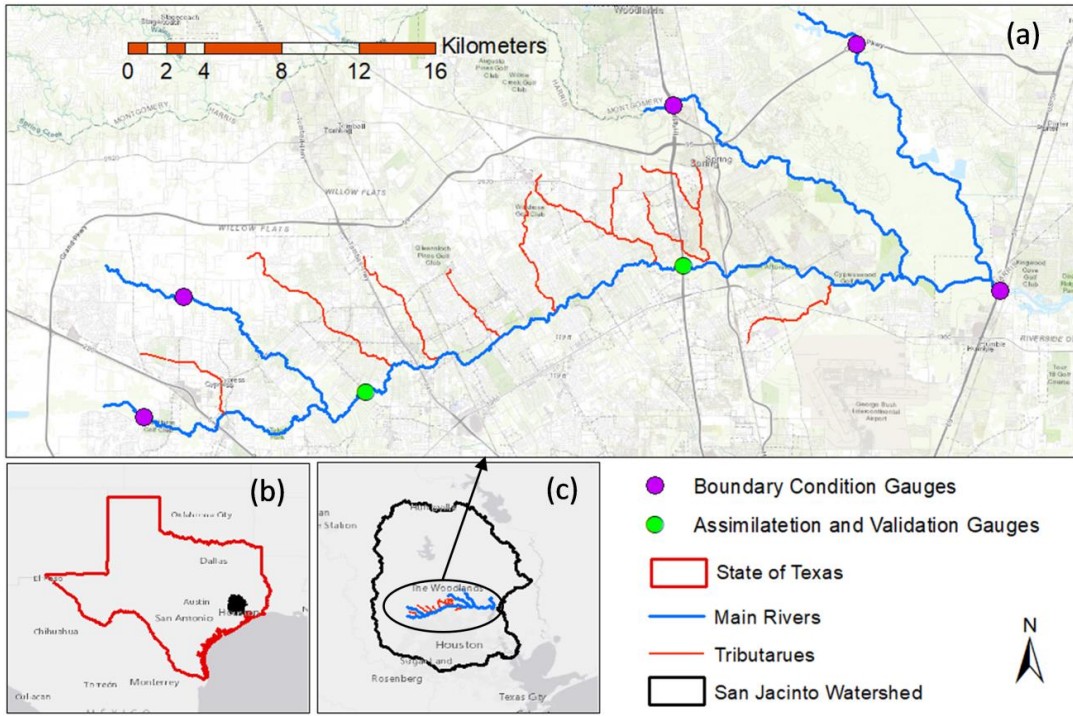


*Figure 1 (a) Study area with all gauges, rivers, and tributaries. (b) Geographic location of San Jacinto Watershed within the state of Texas. (c) Geographic location of the study area within San Jacinto watershed (© NhDplus and USGS).*

## 3. Methods

### 3.1 Flood inundation model

The flood inundation model used in this study is LISFLOOD-FP (Bates and De Roo, 2000), a raster-based 2D hydrodynamic model that simulates the spatiotemporal distribution of water surface elevation over the study area. The model solves the momentum and continuity equations (Saint Venont equations):

$$\frac{\partial Q}{\partial x} + \frac{\partial A}{\partial t} = 0 \qquad (1)$$





$\quad \frac{1}{A}\frac{\partial A}{\partial t} + \frac{1}{A}\frac{\partial (\frac{Q^2}{A})}{\partial x} + g\frac{\partial h}{\partial x} - g(S_0 - S_f) = 0$ $\qquad\qquad$ (2)
where $Q$ is the flow rate at a given cross-section with the area of $A$ in the main channel, $x$ denotes
the location along the channel, t represents time, $S_0$ and $S_f$ are channel bed and friction slopes, and
$g$ is the gravitational acceleration.
We use the sub-gird channel solver, the most recently developed numerical scheme that considers
friction and water slope as well as local acceleration components in the shallow water equations
(Neal et al., 2012). This solver is advantageous for large-scale and efficient modeling as it utilizes
coarse resolution DEMs along with channel width values that are smaller than DEM resolution.
Since DA-hydrodynamic modeling requires hundreds of model simulations, a computationally
intensive operation, this solver helps reduce the computational burden and enables implementing
probabilistic flood inundation mapping within a DA framework. To set up the model, we assume
rectangular cross-section areas and a uniform roughness for both channel and floodplain. Given
the low sensitivity of LISFLOOD-FP to the floodplain roughness (Hall et al., 2005), this parameter
is assumed a constant value. However, the channel roughness is the only model roughness
parameter whose associated uncertainty is accounted for within the assimilation framework. We
also consider the uncertainty of bathymetry by defining an offset parameter that uniformly lowers
the DEM values of the river channels. In addition to model parameters (channel roughness and
bathymetry), the upstream and lateral fluxes entered the river system as the boundary conditions
of the model are other main sources of uncertainty in the assimilation framework.
The upstream boundary conditions are generated from four USGS gauge stations (Figure. 1). To
estimate the lateral fluxes, we calculate the deficit in the system as subtraction of the upstream
from downstream flows and then, distribute the deficit among river tributaries based on their





drainage areas (Please refer to Jafarzadegan et. al (2021) for detailed information about the
calculation of lateral flows in this study area). In section 3.3, we will further discuss the procedure
we used to initialize the model parameters and river boundary conditions.

## 238    3.2 Ensemble Kalman Filter (EnKF)

(Moradkhani et al., 2005b) provided a comprehensive description of the EnKF formulation for
dual estimation of state and parameters in hydrologic models. Here we briefly describe the EnKF
formulation for multivariate assimilation of point source water stage and discharge data into a
hydrodynamic model. For a more effective assimilation proccess, both types of interconnections
between observations, namely spatial correlation of a single observation (discharge or water stage)
among different gauges as well as the correlation between both observations at a single gauge are
taken into account in the EnKF equations. In this study, EnKF is used to simultaneously estimate
model states and parameters. For this purpose, the parameters should be treated similar to the state
variables with a difference that parameter evolution is generated artificially.
Let's assume a DA-hydrodynamic modeling framework with $l$ parameters ($p = 1,2, \dots, l$), $m$ states
($s = 1,2, \dots, m$) and $n$ observations ($j = 1,2, \dots, n$). The following EnKF equations are described
in accordance with the flowchart shown in Figure 2. In the EnKF, parameter samples can be
generated by adding the noise of $\eta_t$ with covariance $\sum_t^\theta$ to the prescribed parameters.
$$\theta_{t+1}^{i-} = \theta_t^{i+} + \tau_t^i \qquad \tau_t^i \sim N(0, \eta_{t+1}) \quad \forall \quad \eta_{t+1} = \sum_{t+1}^\theta \qquad\qquad (3)$$
Using $\theta_{t+1}^{i-}$ and forcing data, a model state ensemble and predictions are generated, respectively.
$$x_{t+1}^{i-} = f\left(x_t^{i+}, u_t^i, \theta_{t+1}^{i-}\right) + \omega_t^i \quad \omega_t^i \sim N(0, Q_t) \quad \forall \quad Q_t = \sum_t^x \qquad\qquad (4)$$
$$\hat{y}_{t+1}^i = h\left(x_{t+1}^{i-}, \theta_{t+1}^{i-}\right) + v_{t+1}^i \qquad v_{t+1}^i \sim N(0, R_{t+1}) \quad \forall \quad R_{t+1} = \sum_{t+1}^y \qquad\qquad (5)$$





where $x_t$, $u_t$, $\theta_t$ and $y_t$ are the vector of the uncertain state variables, forcing data, model
parameters and observation data at time step $t$, respectively. $\omega_t$ represents the model errors due to
the imperfect model, and $\nu_t$ is the measurement error. Most often, $\omega_t$ and $\nu_t$ are assumed to be
white noises with mean zero and covariance $Q_t$ and $R_t$, respectively. In addition, the two noises
$\omega_t$ and $\nu_t$ are assumed to be independent.
Then we update the parameter ensemble members using the standard Kalman filter equation:
$$\theta_{t+1}^{i+} = \theta_{t+1}^{i-} + K_{t+1}^{\theta}\left(y_{t+1}^i - \hat{y}_{t+1}^i\right) \tag{6}$$
where $K_{t+1}^{\theta} \in \mathbb{R}^{l\times n}$ is the Kalman gain matrix for correcting the parameter trajectories and is
obtained by:
$$K_{t+1}^{\theta} = \Sigma_{t+1}^{\theta y}\left[\Sigma_{t+1}^{yy} + R_{t+1}^{'}\right]^{-1} \tag{7}$$
where $\Sigma_{t+1}^{\theta y} \in \mathbb{R}^{l\times n}$ is the cross-covariance matrix of parameter ensemble and prediction ensemble
(Eq. 6). Unlike other studies, and for more realistic characterization of observation and model
errors here the correlation between the errors associated with $n$ observation data are accounted for
during the assimilation process. Therefore, the covariance matrix $R'_t \in \mathbb{R}^{n\times n}$ is a nonzero matrix,
such that the values in the diagonal represent the error associated with each observation data and
all elements lower/upper the main diagonal denote the cross covariance between different
observations (Eq. 7). $\Sigma_t^{yy} \in \mathbb{R}^{n\times n}$ is also a similar covariance matrix with the inclusion of error
correlation between the model simulations  (Eq. 8).
$$\Sigma_{t+1}^{\theta y}(p,j) = \frac{1}{N}\sum_{i=1}^{N}\left[\left(\theta_{t+1}^{i-}(p) - E[\theta_{t+1}^-(p)]\right)\left(\hat{y}_{t+1}^i(j) - E[\hat{y}_{t+1}(j)]\right)\right] \tag{8}$$
$$R_{t+1}^{'}(j,j') = \begin{cases} R_{t+1} & j = j' \\ \frac{1}{N}\sum_{i=1}^{N}\left[\left(y_{t+1}^i(j) - E[y_{t+1}(j)]\right)\left(y_{t+1}^i(j') - E[y_{t+1}(j')]\right)\right] & j \neq j' \end{cases} \tag{9}$$



$\quad \Sigma_{t+1}^{yy}(j,j') = \frac{1}{N}\sum_{i=1}^{N}\left[\left(\hat{y}_{t+1}^{i}(j) - E[\hat{y}_{t+1}(j)]\right)\left(\hat{y}_{t+1}^{i}(j') - E[\hat{y}_{t+1}(j')]\right)\right]$ (10)
$\quad E[\theta_{t+1}^{-}] = \frac{1}{N}\sum_{i=1}^{N}\theta_{t+1}^{i-}$ (11)
$\quad E[\hat{y}_{t+1}] = \frac{1}{N}\sum_{i=1}^{N}\hat{y}_{t+1}^{i}$ (12)
Now using the updated parameter, the new model state trajectories (state forecasts) and prediction
trajectories are generated:
$\quad x_{t+1}^{i-} = f\left(x_{t}^{i+}, u_{t}^{i}, \theta_{t+1}^{i+}\right) + \omega_{t}^{i} \quad \omega_{t}^{i} \sim N(0, \Sigma_{t}^{x}) \quad \forall \quad Q_{t} = \Sigma_{t+1}^{x}$ (13)
$\quad \hat{y}_{t+1}^{i} = h\left(x_{t+1}^{i-}, \theta_{t+1}^{i+}\right) + v_{t+1}^{i} \quad v_{t+1}^{i} \sim N\left(0, \Sigma_{t+1}^{y}\right) \quad \forall \quad R_{t+1} = \Sigma_{t+1}^{y}$ (14)
Model states ensemble is similarly updated as follows:
$\quad x_{t+1}^{i+} = x_{t+1}^{i-} + K_{t+1}^{x}\left(y_{t+1}^{i} - \hat{y}_{t+1}^{i}\right)$ (15)
$\quad y_{t+1}^{i} = y_{t+1}^{i} + v_{t+1}^{i} \quad v_{t+1}^{i} \sim N(0, R_{t+1}) \quad \forall \quad R_{t+1} = \Sigma_{t+1}^{y}$ (16)
where $K_{t+1}^{x} \in \mathbb{R}^{m \times n}$ is the Kalman gain for correcting the state trajectories and is obtained by:
$\quad K_{t+1}^{x} = \Sigma_{t+1}^{xy}\left[\Sigma_{t+1}^{yy} + R_{t+1}'\right]^{-1}$ (17)
where $\Sigma_{t+1}^{xy} \in \mathbb{R}^{m \times n}$ is the cross-covariance matrix of states ensemble and prediction ensemble
(Eq. 16).
$\quad \Sigma_{t+1}^{xy}(s,j) = \frac{1}{N}\sum_{i=1}^{N}\left[\left(x_{t+1}^{i-}(s) - E[x_{t+1}^{-}(s)]\right)\left(\hat{y}_{t+1}^{i}(j) - E[\hat{y}_{t+1}(j)]\right)\right]$ (18)
$\quad E[x_{t+1}^{-}] = \frac{1}{N}\sum_{i=1}^{N}x_{t+1}^{i-}$ (19)
In this study the water depth along the channel is the only state variable ($m$=1). The channel
roughness and bathymetry are two model parameters ($l$=2) and three point source observations





including water discharge at gauge 1 and 2 as well as water stage at gauge 2 (n=3) are assimilated
into the LISFLOOD-FP model (Table 1). Therefore, the Kalman gains used to update the model
parameters and states (Eqs 5 and 15) are $2 \times 3$ and $1 \times 3$ matrices that take advantage of a
multivariate point source assimilation while considering the downstream correlation between
discharge observations and the correlation between water stage and discharge at gauge 2. **3.3.**

## 299    **Experimental design**

The ultimate goal of this study is to simulate the Hurricane Harvey flood and generate probabilistic
flood inundation maps through the DA-hydrodynamic modeling framework. Figure. 1 illustrates
the flowchart of the proposed probabilistic flood inundation mapping approach. In this study, the
EnKF is performed based on an ensemble size of 100. The boundary conditions including four
upstream flows, seven lateral fluxes, and downstream flows are perturbed with adding white noises
sampled from a normal distribution with a mean zero and relative error of 20%. The errors are
assumed heteroscedastic meaning that their values are proportional to the flow magnitude. To
characterize uncertainty in the initial condition, namely water depth, we add a white noise with a
mean zero and standard deviation of 1 meter. In this study, using the proposed EnKF-based
multivariate assimilation approach, three point-scale observations, i.e., discharge at USGS gauges
1 and 2, as well as water stage at gauge 2, are incorporated into the LISFLOOD-FP model to rectify
its state variables and parameters, and hence provide more accurate and reliable flood inundation
maps. All these three observations are perturbed by adding a normally distributed white noise with
a mean zero and a relative error of 20%.  First, the LISFLOOD-FP model is forced with the
upstream, downstream and lateral flow ensembles. To initialize the state variables in the system,
the simulated water depth values at the ending day of the warm-up period (the initial condition for
the first day of the model simulation) are perturbed with adding a white noise with a mean zero





and standard deviation of 1 meter. The model parameters (i.e., channel roughness and bathymetry)
are initialized using the Latin Hypercube Sampling method and evolved during the assimilation
process. The ensemble of water depth values predicted by the model for the next time step together
with observations, namely water stage and discharge at gauges are used in the multivariate Kalman
equation to update the model parameters. The LISFLOOD-FP model is run for the second time
with the updated parameters and the second multivariate Kalman equation uses the predicted water
depth with observations to update the ensemble of water depth in the system. The ensemble of
updated water depth (state), bathymetry, and channel roughness (parameters) will be used within
the LISFLOOD-FP to predict an ensemble of water depth for the next time step. The predicted
water depth is simply converted to a probabilistic flood inundation map. Using this data
assimilation framework, we can generate 1-day forecast of probabilistic flood inundation maps
which would be highly beneficial for real-time flood warning and decision making. It is worth
mentioning that the forecasted probabilistic maps account for different sources of uncertainty
including the forcing data (boundary condition flows), model parameters (channel roughness and
bathymetry), and initial conditions (water depth).
The simulation period of the LISFLOOD-FP model is set up for 45 days from July-30-2017 to
Sep-12-2017 and the entire month of July is used as a warm-up period. The water depth generated
for the end of July will be used as the initial condition of the model. To account for the uncertainty
of channel roughness and bathymetry, we sample them from uniform distributions ranging from
[0,0.1] and [39,42] m, respectively. The bathymetry parameter is the elevation of the channel bed
at the upper location of the channel. The offset parameter is calculated by subtracting this value
from DEM at the upper location. Then, the bathymetry vector that includes the channel bed
elevation for all channel cells is generated by subtracting the offset from DEM values along the





channel. It should be noted that the range of uniform distribution is chosen based on previous
studies (Aronica et al., 2002b; Bales and Wagner, 2009; Di Baldassarre et al., 2009; Horritt, 2006;
Pappenberger et al., 2008), expert judgment, and trial-and-error.

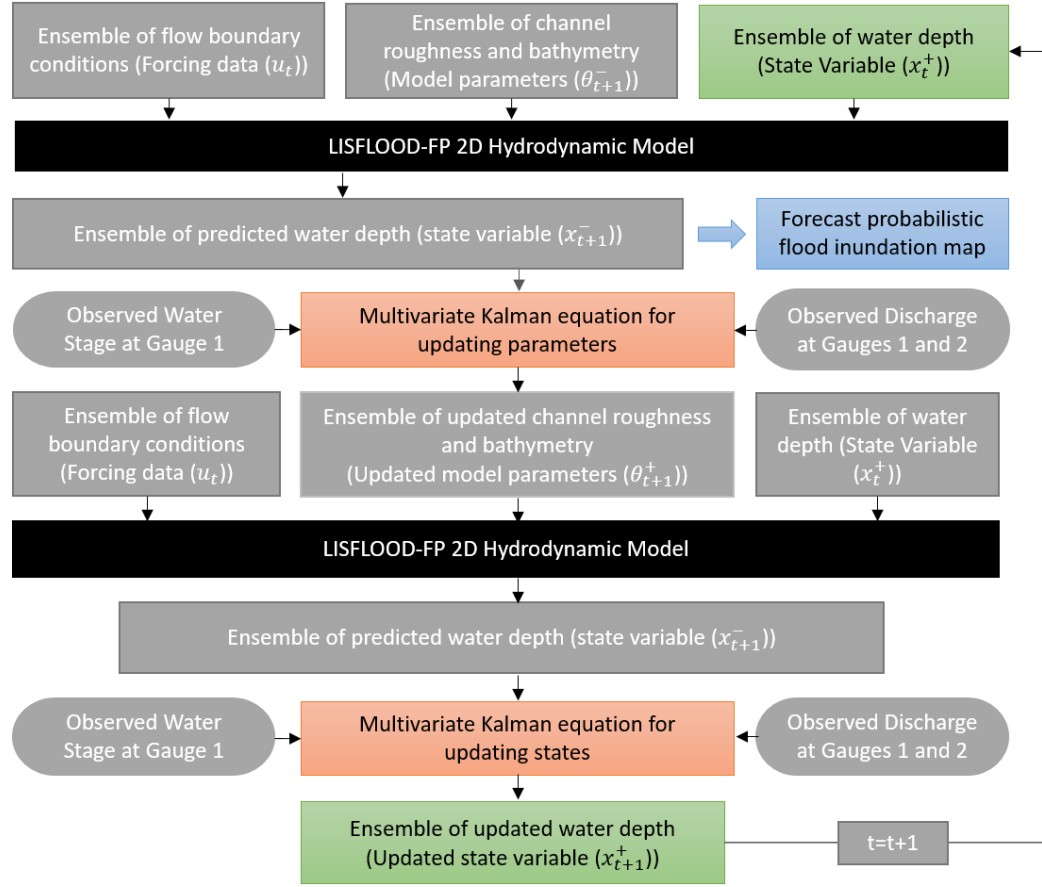


*Figure 2. Schematic of the DA-hydrodynamic modeling framework for real-time probabilistic*
*flood inundation mapping. The green boxes represent the state variables where their updated*
*values are fed into the LISFLOOD-FP model and provide a probabilistic flood inundation map*
*at the forecast mode (blue box). The black boxes highlight the physical model and the orange*
*boxes represent the Kalman equations used for updating the parameter and state variables by*
*the EnKF.*
To assess the effectiveness and robustness of the proposed assimilation framework for
probabilistic flood inundation mapping, we design three different experiments. First, an open-loop



(OL) simulation is established where the model is run without assimilation. In the second
experiment, we perform DA-hydrodynamic modeling on a synthetic case study where we assume
the model is perfect and has no error. In this approach, we set the model parameters (channel
roughness and bathymetry), initial state (water depth) and boundary condition flows to fixed values
and run the model to generate discharge and water surface elevation across the gauges within the
study area. These predicted values are assumed as benchmark observations. This synthetic analysis
ensures that the assimilation process performs well and the model parameters end up converging
to predefined values. In the next step, we implement the proposed assimilation framework on a
real case study where the observed discharge and water surface elevation data that are recorded
from the USGS gauges during Hurricane Harvey, are assimilated into the model. Considering the
severe flood condition during the Hurricane, we aim to investigate the extent to which the
multivariate DA-Hydrodynamic modeling framework improves the model simulation and flood
inundation mapping skill.
**3.4 Validation strategy**
As mentioned before, the convergence of uncertain model parameters toward truth in the
synthetic experiment demonstrates the performance of DA-hydrodynamic modeling framework.
To provide a robust analysis of each assimilation run, it is necessary to assess the model
performance through multiple deterministic (KGE and RMSE) and probabilistic (NRR and
Reliability) measures. The summary of performance measures used in this study is tabulated in
Table 1.






Table 1: Summary of performance measures used in this study

| Performance Measure | Mathematical Representation |
|---|---|
| Kling-Gupta Efficiency (KGE) | $1 - \sqrt{\left(\left(\frac{\text{Cov}_{y_t y'_t}}{\sigma \sigma'}\right) - 1\right)^2 + \left(\left(\frac{\sigma'}{\sigma}\right) - 1\right)^2 + \left(\left(\frac{\mu'}{\mu}\right) - 1\right)^2}$ |
| Root Mean Square Error (RMSE) | $\sqrt{\frac{1}{T}\sum_{t=1}^{T}(y'_t - y_t)^2}$ |
| Normalized Root-Mean-Square Error Ratio (NRR) | $\sqrt{\frac{1}{T}\sum_{t=1}^{T}(y_t - \overline{y'_{\blacksquare,t}})^2} \times \left(\frac{1}{T}\left\{\sum_{t=1}^{T}\sqrt{\frac{1}{T}\left[\sum_{t=1}^{T}(y_t - \overline{y'_{\blacksquare,t}})^2\right]}\right\}\sqrt{\frac{N+1}{2N}}\right)^{-1}$ |
| Reliability | $1 - \frac{2}{T}\sum_{t=1}^{T}\left|\frac{Z_t}{T} - U_t\right|$ |


$y_t$ and $y'_t$ are the observed and simulated values, respectively. The Kling–Gupta Efficiency (KGE)
varies from $-\infty$ to 1, such that a value of 1 indicates a perfect fit between observed and simulated
values. The pairs of $(\mu, \sigma)$ and $(\mu', \sigma')$ represent the first two statistical moments (means and
standard deviations) of $y_t$ and $y'_t$, respectively. Root mean squared error (RMSE) is the square
root of the mean of the square of all of the error between the predicted and observed values.
NRR (DeChant and Moradkhani, 2012) is calculated to measure the ensemble spread and assess
how confidently the ensemble mean is statistically distinguishable from the ensemble spread.
Reliability (Renard et al., 2010) is a measure of the fit of the Q-Q quantile plot to a uniform. A
value of 1 is exactly uniform and a value of 0 is the farthest possibility from uniform. For the
description of the $z_t$ and $U_t$ calculation, we refer the readers to Renard et al. (2010).


The above four performance measures assess the dynamic behavior of DA-hydrodynamic
modeling framework at two specific points. Moreover, to spatially evaluate the behavior of the
proposed framework, we compare the maximum probabilistic flood inundation maps (union of
probabilistic maps over the simulation period) with the observed floodplain map delineated
aftermath of Harvey. The Receiver Operating Characteristic (ROC) graph is a common tool for
validating probabilistic classifiers (Fawcett, 2006). Consider a deterministic flood map as a binary
map where one and zero represent flooded and non-flooded cells, respectively. First, a threshold
in the range of [0,1] is used to convert the probabilistic map to a binary deterministic map. This
means all cells with the probability of inundation less than a given threshold are converted to zero
and other cells are set to one. The binary map is compared with the reference map and the rate of
true positive (rtp) and false positive (rfp) are calculated using Equations 7 and 8 (Jafarzadegan and
Merwade, 2017):
$$rtp = \frac{True\ positive\ instances}{total\ positives} \qquad (7)$$
$$rfp = \frac{False\ positive\ instances}{total\ negative} \qquad (8)$$
where true and false positive instances represent the total number of flooded cells in the reference
map that are predicted as flood and non-flooded cells, respectively. Total positives and negatives
are total flooded and non-flooded cells in the reference map. This process is repeated and a set of
points (rfp.rtp) are generated corresponding to different thresholds. The ROC graph connects the
points in the rfp-rtp space and the area under the curve (AUC) represents the performance of the
probabilistic classifier (Fawcett, 2006). In this study, we use AUC to compare the performance of
OL simulation with the EnKF for probabilistic flood inundation mapping. In addition, we calculate



the Underprediction and Overprediction Flood Indices (UFI and OFI) introduced by Jafarzadegan
et al., (2018) for comparing probabilistic flood maps against deterministic reference maps:
$$UFI = \frac{\sum_{i=1}^{N}(1-P_i)}{N} \times 100 \qquad i \in F \qquad\qquad (9)$$
$$OFI = \frac{\sum_{i=1}^{M}(P_j)}{M} \times 100 \qquad j \in NF \qquad\qquad (10)$$
where F and NF denote the flooded and non-flooded regions in the reference map, and i and j are
indicators of cells located within these regions. N and M are the total number of cells in the F and
NF regions and $P_i$, $P_j$ denote the probability of inundation for cells i and j derived from the
probabilistic flood maps.
## 4. Results
### 4.1 Synthetic Case Study
We conduct the synthetic experiment to ensure the usefulness and effectiveness of the proposed
DA-hydrodynamic modeling framework. Figure 3 presents uncertainty bound evolution of the
parameters in the LISFLOOD-FP model (i.e., channel roughness and bathymetry) for 45 days
assimilation of synthetic observations (i.e., discharge at gauges 1 and 2 and water stage at gauge
2). The shaded areas correspond to 95, 75, 68, and 10 percentile predictive intervals, and the black
stars at the end of each parameter subplot represent the true parameter values. As seen both
parameters converge smoothly to the certain region in parameter space where the uncertainty
bounds stabilize. While the uncertainty bound associated with the bathymetry becomes stabilized
at the early stage of the assimilation process, for the channel roughness, the uncertainty bound gets
stabilized toward the end of the assimilation period. It is also evident from Figure 3 that the





bathymetry is a more identifiable parameter as it shows the fastest convergence with a minimum
degree of uncertainty. However, the channel roughness is less identifiable with the slowest
convergence. The scatter plots illustrate the evolution of parameter space at six different time
segments. The first day (t=1) includes all 100 ensemble members of parameters and day 30
corresponds to the highest discharge and water stage of flooding when the model parameters reach
the highest improvement and get closer to the true value. Figure 3 shows that both model
parameters are converging toward the true values as the assimilation proceeds. This indicates the
efficacy and usefulness of the proposed DA-hydrodynamic modeling framework developed in this
study.



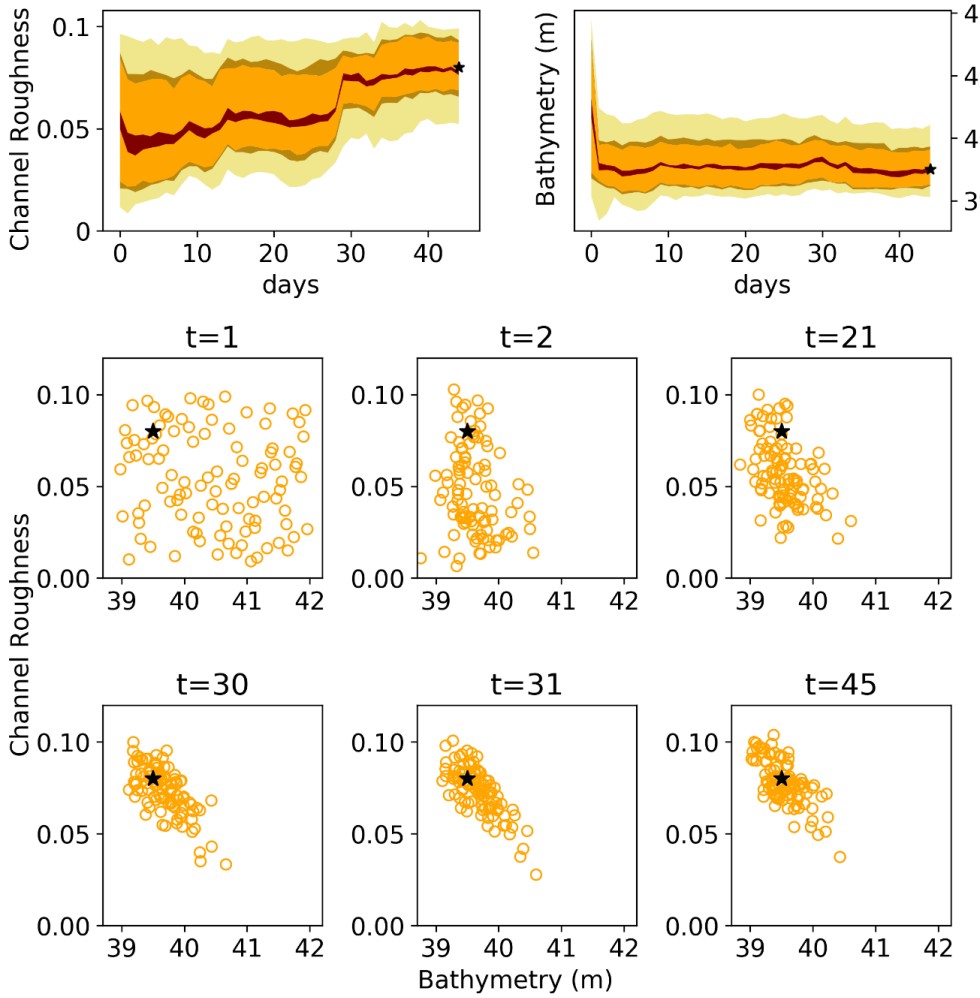


*Figure 3. Temporal evolution of the LISFLOOD parameters for the synthetic experiment during Hurricane Harvey using the EnKF. (a) Temporal evolution of model parameter predictive intervals corresponding to 95, 75, 68, and 10 percentile (b) Temporal evolution of particle positions in the model parameter space at six different days during the Hurricane.*

## 4.2 Real Case Study


In the real experiment, we assimilate the discharge and water stage readings from two internal
USGS gauges into the LISFLOOD-FP model. We also run the OL simulation and calculate the
ensemble mean to predict the discharge and water stage at these two gauges. Figure 4 presents a





comparison of simulated discharge (Figures 4a, 4b) and water stage (Figures 4c, 4d) with

observations using both OL and our EnKF-based approach. Figures 4a and 4c are the prior

estimates of discharge and water stage, while Figures 4b and 4d show their posteriors which reflect

the updated variables after assimilating the observations into the model. It is worth mentioning

that although priors represent the results before assimilating new observations into the model, their

values are dependent on the initial conditions updated from observations in the previous time step.

In this study, since forecasting (1-day lead time) is the main objective of DA-hydrodynamic

modeling framework, we specifically focus on behavior of priors. As can be seen, the simulated

peak discharge by the OL is highly overestimated by around 200 cms while assimilating the

observations improve the results so that their difference with observation is less than 50 cms at the

peak of the flood (KGE =0.76 and RMSE=40.9 cms)). In contrast, the simulated water stage in

Figures 4c and 4d are underestimated by OL by around 2 meters at the peak. Using the developed

approach raises the peak of water stage at peak and reduces the errors significantly (KGE=0.96

and RMSE=0.5). The accurate estimates of prior discharge and water stage confirm the

applicability of the proposed assimilation framework in forecast mode when real-time flood

warning and decision making is the priority. The NRR measure for the prior discharge and water

stage are 1.17 and 0.65 showing that the uncertainty bound is underestimated and overestimated,

respectively. The reliability of both variables is above 70 percent since the uncertainty bounds

encompass the observations for almost the entire simulation period.

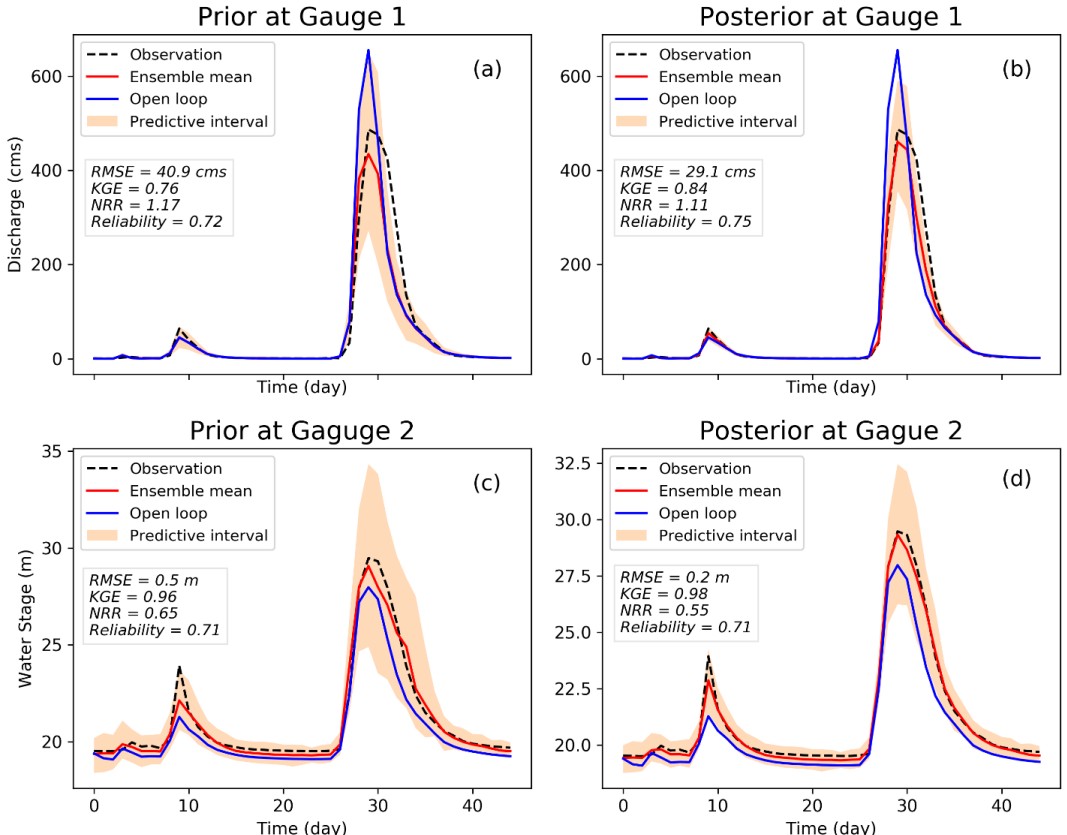

465

*Figure 4 Simulation results of LISFLOOD-FP for the real experiment during Hurricane Harvey*
*using the EnKF and open-loop. (a) Prior simulated discharge at gauge 1 (b) Posterior simulated*
*discharge at gauge 1 (c) Prior simulated water stage at gauge 2 (d) Posterior simulated water*
*stage at gauge 2. The shaded areas represent the predictive interval of simulated discharge and*
*water stage by EnKF.*

Figure 5 illustrates the prior and posterior distributions of discharge and water stage in the

beginning, peak, and ending days of Hurricane Harvey flood. In all three days, the uncertainty

bounds of both discharge and water stage are narrowed down by assimilating the observations so

that posterior distributions are more precise compared to the priors. In the beginning and ending

days (Aug 26 and Sep 1) the mean of prior distributions is substantially shifted toward truth in the

posterior distributions. Figure 5 reveals that our developed approach provides more accurate and





reliable posterior discharge and water stage distributions compared to prior distributions where the
simulations are either overestimated or underestimated. It is noted that, on August 28 (day of flood
peak), although the prior distributions accurately represent the observation, they have wide
uncertainty bound. After correcting/updating the model state variables and parameters, as posterior
distributions show, the uncertainty bound is reduced while the ensemble mean remains closer to
the observation.

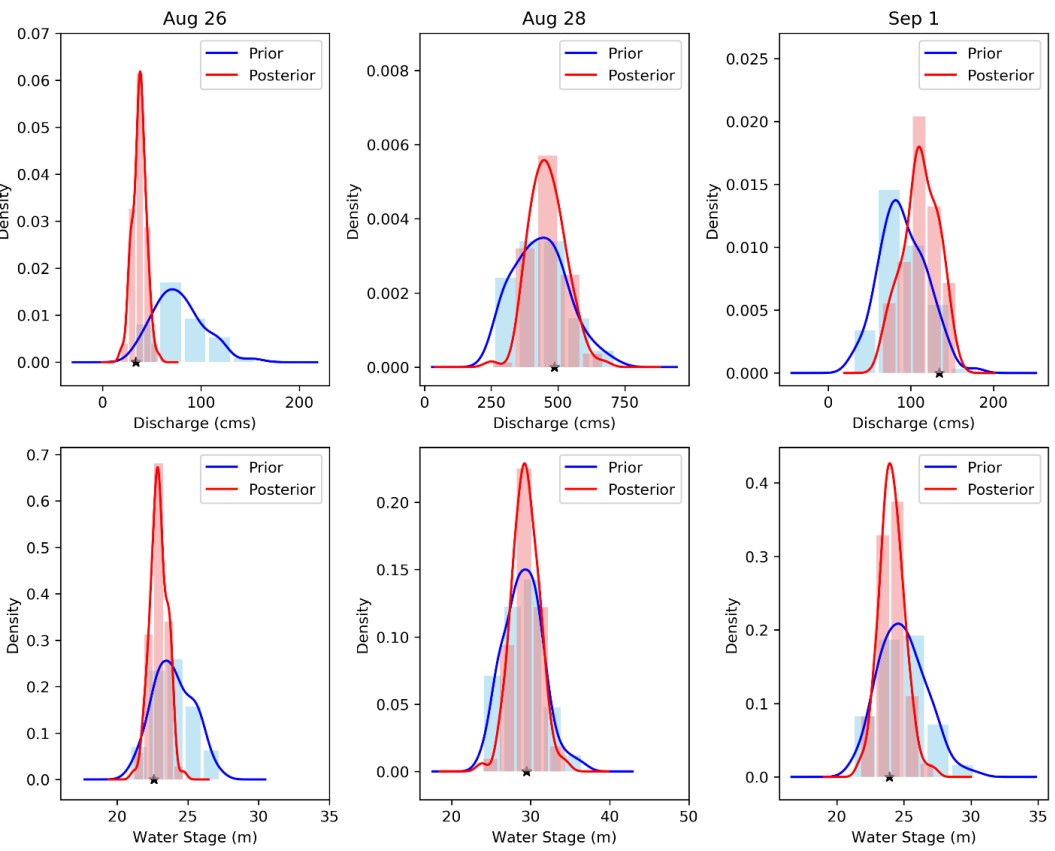


*Figure 5. Prior and posterior distribution of discharge (a,b,c) and water stage (c,d,f) at the*
*beginning (Aug 26), peak (Aug 28), and ending (Sep1) days of Hurricane Harvey using the*
*EnKF*



## 4.3 Probabilistic Flood Inundation Mapping

In this study, we propose a DA-hydrodynamic modeling framework to account for the uncertainties involved in flood modeling and generate real-time probabilistic flood inundation maps. Since the majority of flooding conditions occurred within 6 days from August 27-Sep 1, we display the spatial distribution of water depth in this period and provide probabilistic flood inundation maps using both OL and our developed approach (see Figures 6 and 7). Figure 6 represents the first three days of Harvey which corresponds to the upper limb of the flood hydrograph. On August 27, the major difference between the OL and EnKF appears in the regions around the upstream of the lower channel where the EnKF provides a more reliable inundated area. Moving toward the peak of flood on Aug 29, the OL generates a large region of uncertain cells around the banks of the upper channel while both the extent and density of uncertain values in the probabilistic maps generated by the EnKF is smaller during the peak of Harvey.



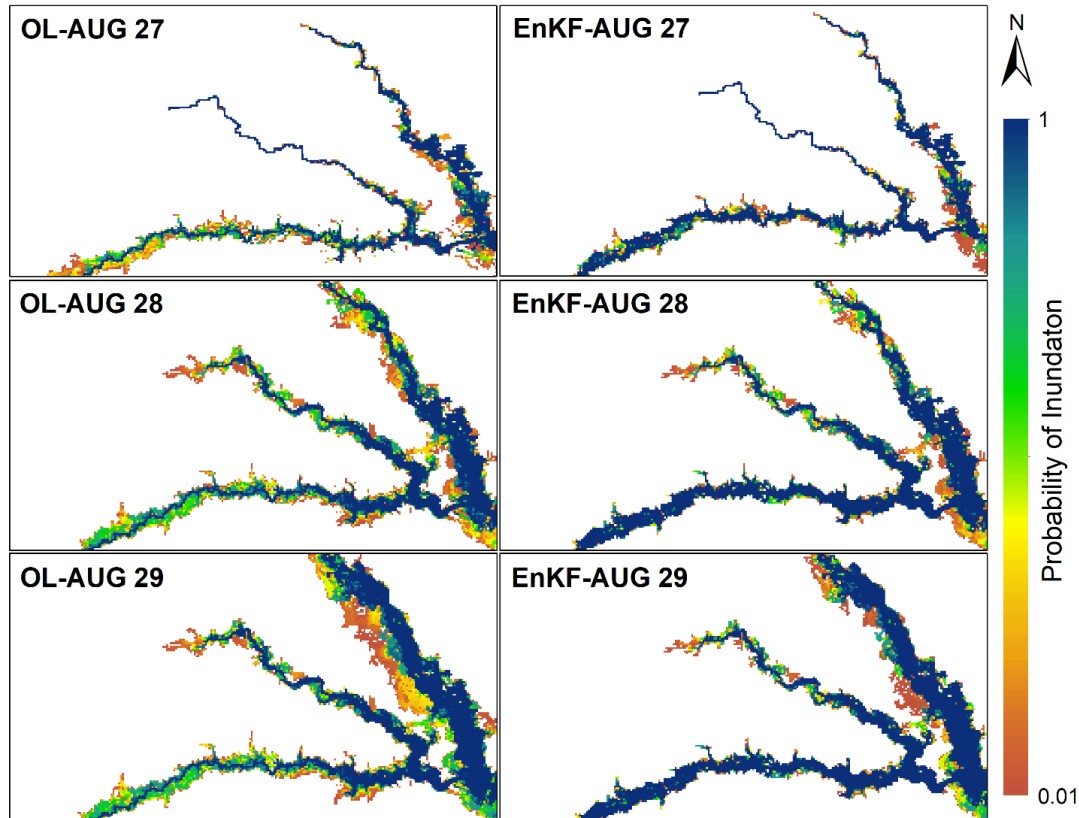

500

*Figure 6 Probabilistic flood inundation maps generated by OL and EnKF techniques to simulate the upper limb of Harvey flood hydrograph from Aug 27 to Aug 29.*

Figure 7 shows the probabilistic inundation areas in the last three days corresponding to the lower limb of the flood hydrograph. In this figure, the discrepancies between the OL and EnKF flood maps increase showing that performing DA is more effective in improving the inundation mapping skill from peak to ending point of the flood hydrograph. A large number of inundated cells generated by the OL are vanished after the peak of Harvey which results in a set of scattered discontinuous maps in Aug 31 and Sep 1. On the other hand, the probabilistic maps generated by the EnKF maintain their continuous shapes so that the probability of inundation is reduced without changing the extent. The merit of the EnKF in improving the flood inundation areas at the lower





limb of the flood hydrograph agrees with results in Figures 4c and 4d where the EnKF widens the
simulated water stage hydrographs and removes the lag difference that exists between the open-
loop and observations.

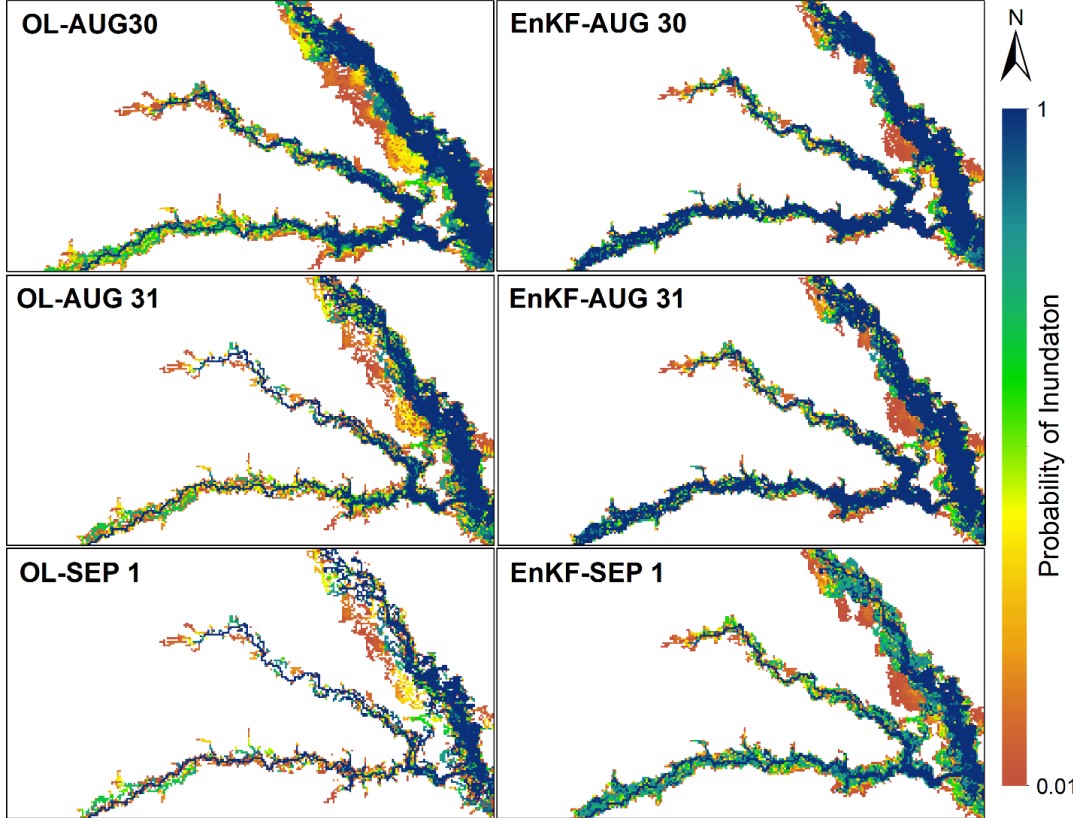


*Figure 7 Probabilistic flood inundation maps generated by OL and EnKF techniques to simulate*
*the lower limb of Harvey flood hydrograph from Aug 30 to Sep 1.*
Finally, to quantify the performance of EnKF and OL for generating a spatial distribution of water
depth over the domain, we illustrate the ROC graphs, the AUC values, and Fit indices in Figure 8.
To calculate these measures, we ignore the temporal distributions and only report the maximum
inundation maps that represent the union of flooded areas over the entire period of Harvey.
Comparing the EnKF and OL in Figure 8.a, the EnKF line (blue) is closer to the northwest of the




rfp-rtp space where its AUC is 5% higher than the OL approach. In Figure 8.b, each point
represents the Fit indices for the OL and the EnKF approaches corresponding to a given threshold.
Using hundred number of thresholds that each ranging from [0.01,1], the probabilistic maps are
converted to 100 deterministic maps and the Fit indices are calculated. The position of scatters
above the dash line confirms the EnKF outperforms the OL. In addition to these measures, the
[UFI, OFI] indices calculated for OL and EnKF approaches are [30.3, 0.26] %, and [23.4, 0.4]%
respectively. The low values of OFI for both approaches ($< 1\%$) show that the simulations mostly
underestimate the flood inundation areas. In addition, comparing the indices of both approaches
reveal that the EnKF reduces the overall underestimation by around 7%.

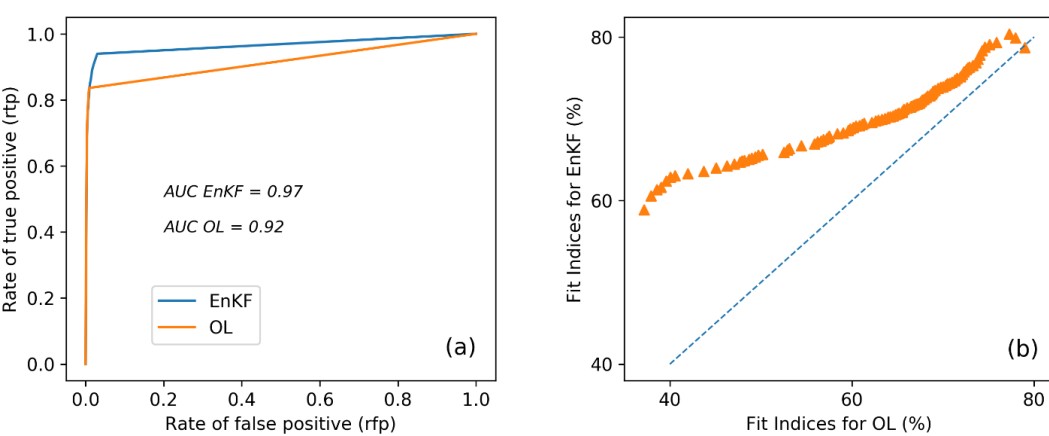


*Figure 8 The Receiver Operating Curves (ROC) indicating the performance of OL and EnKF*
*techniques for probabilistic flood inundation mapping*
**5. Discussion and Conclusions**
The main motivation in this study is to propose a DA-hydrodynamic modeling framework for real-
time probabilistic flood inundation mapping. Considering the coarse spatiotemporal resolution of
satellite data for capturing the water surface elevation, assimilating them into the hydrodynamic



models may not be a practical solution for an upcoming flood event. On the other hand, the
availability of daily discharge and water surface elevation data at gauge stations is a great
opportunity to establish a multivariate DA-hydrodynamic modeling framework that updates the
initial condition of modeling at daily scale and forecast the flood inundation areas at 1 day lead
time.  Here, we used the EnKF data assimilation method in conjunction with a hydrodynamic
model to account for different sources of uncertainties involved in different layers of model
simulations, including the boundary conditions, model parameters, and initial condition,  and
generate real-time probabilistic flood inundation maps . To further enhance the performance of the
developed framework, the discharge and water stage at two different gauges are simultaneously
assimilated into the LISFLOOD-FP model. The multivariate EnKF approach considers the
correlation between discharge at two gauges and between discharge and water surface elevation at
one gauge using a modified covariance matrix and Kalman gain equation.
In the synthetic experiment, we examined the convergence of model parameters toward truth and
found that the proposed DA-hydrodynamic modeling framework can be successfully used to
improve the accuracy and reliability of model predictions while accounting for uncertainties
associated with model parameters. The channel roughness coefficient varied more  rapidly than
the bathymetry during the temporal evolutions of these parameters showing the better
idemtifiability of this parameter. The validation results of the real experiment revealed that the
assimilation with the EnKF approach improves the model predictions at across temporal and
spatial scales (i.e., discharge and water stage time series at gauges and flood maps showing the
maximum water depth over the simulation period). These improvements are more pronounced
during the falling limb of the flood hydrograph where the EnKF widens the simulated hydrograph
and removes the existing lag compared to the observations. Similarly, the simulated flood





inundation maps confirm that the OL provides discontinuous scattered maps during the flood
recession period while the EnKF provides a more accurate representation of the inundation areas.
The validation results also demonstrate that the EnKF reduces the underestimation by 7% and
outperformed the OL approach by around 5% for probabilistic flood inundation mapping.
To simulate flood hazards during the emergency of an upcoming flood event, using an efficient
flood modeling framework is of paramount importance. However, a simplified model setup (i.e.
using coarse resolution DEM, assuming uniform roughness coefficient for channel and floodplain,
estimating bathymetry by lowering DEM with one parameter) for efficient flood modeling is prone
to losing accuracy. Particularly, for an extreme flooding condition such as Hurricane Harvey, the
simplified modeling may pose significant errors. The results obtained from the simulation of the
real experiment demonstrated that despite using a simplified efficient modeling setup, we can still
simulate the discharge, water stage, and inundation areas for an extreme flood event with
acceptable accuracy while accounting for uncertainties involved in model predictions. This shows
that assimilating the gauge data into a simplified model setup improves the accuracy, and provides
an efficient probabilistic framework for real-time flood inundation mapping that considers
potential sources of uncertainties in different layers of modeling.
The time dependency that exists between the upstream and downstream gauges along a channel
can affect the performance of multivariate assimilation with those gauges. For future studies, using
a more advanced DA technique that fully characterizes the model structural uncertainty
(Abbaszadeh et al., 2019), and considering the time lag dependency between multiple gauges can
improve the performance of modeling and provide more realistic assimilation of the hydrodynamic
models. Finally, proposing a DA-hydrodynamic modeling framework that considers the DEM and



channel width uncertainty can provide a more comprehensive uncertainty quantification for
probabilistic flood inundation mapping in future studies.

**Data availability**

All the data used in this study, including the gauge streamflow and water stage data and the DEMs,
are publicly available from the USGS website and National Elevation Dataset (NED). The
reference flood maps provided for Hurricane Harvey is available from the USGS report at
https://pubs.usgs.gov/sir/2018/5070/sir20185070.pdf.

**Author contribution**

KJ and PA designed the synthetic and real experiments. KJ developed, set up, evaluated and
implemented the DA-hydrodynamic modeling framework for both experiments. PA provided
inputs on the assimilation part. KJ wrote the first draft of the manuscript.HR and PA edited the
manuscript.

**Competing interests**

The authors declare that they have no conflict of interest.

**Acknowledgment**

Partial financial support for this study was provided by the USACE contract #W912HZ2020055.
We would like to thank the anonymous reviewers for their constructuve comments on the original
version of the manuscript.




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

data-driven method for dealing with model structural error in soil moisture data
assimilation. Adv. Water Resour. 132, 103407.




