# Peer review of "Sequential Data Assimilation for Real-Time Probabilistic"

_Hydrology and Earth System Sciences, 2021_

## Referee Comment (RC1)

In this study, the authors present a Data Assimilation (DA)-hydrodynamic modeling framework where multiple gauge observations are integrated into the LISFLOOD-FP model to improve the performance of flood inundation mapping. The results indicate that the multivariate assimilation of point-source observations into hydrodynamic models can improve the accuracy and reliability of probabilistic flood inundation mapping by 5-7% while it also provides the basis for sequential updating and real-time flood inundation mapping. This paper is well written, well-organized and has practical meaning for inundation mapping, risk analysis and decision making. I have some minor comments for the authors to improve their paper.

Line 168: Xu et al., 2017 is used as a reference. However, this paper is not listed in the bibliography.

The text (e.g. names of counties and states) in Figure 1 is blurred. Please improve it.

Line 292-295: The channel roughness and bathymetry are two model parameters (l=2) and three point source observations including water discharge at gauge 1 and 2 as well as water stage at gauge 2 (n=3) are assimilated into the LISFLOOD-FP model (Table 1).

In Table 1 (Line 374), the summary of performance measures used in this study is presented instead of the assimilated observations. Please correct it.

In addition, the water discharge at gauge 1 and 2 as well as water stage at gauge 2 (n=3) are assimilated. The authors could also assimilate the water discharge and the water stage at gauge 1, 2 and 3 to further improve the performance. Is the data available?

Figures 6 & 7: the two figures presented the comparison between OL and EnKF. It would be better to add the real inundation map in the third column of the figures.

Line 542-545: Here, we used the EnKF data assimilation method in conjunction with a hydrodynamic model to account for different sources of uncertainties involved in different layers of model simulations, including the boundary conditions, model parameters, and initial condition, and generate real-time probabilistic flood inundation maps.
It is nice to consider different sources of uncertainty here. How the uncertainty fluctuates with ensemble size and error settings?

Figure 4: it would be better to add the predictive interval of open loop.

Line 112: the following paper may be a good reference for the SMAP soil moisture data assimilation.
Xu et al. Continental drought monitoring using satellite soil moisture, data assimilation and an integrated drought index. Remote Sensing of Environment, 2020, 250:112028.

---

## Referee Comment (RC2)

[revised manuscript text omitted]

$$\Sigma^{yy}_{t+1}(j,j') = \frac{1}{N}\Sigma^{N}_{i=1}\left[(\hat{y}^i_{t+1}(j) - E[\hat{y}_{t+1}(j)])(\hat{y}^i_{t+1}(j') - E[\hat{y}_{t+1}(j')])\right] \tag{10}$$

$$E[\theta^-_{t+1}] = \frac{1}{N}\Sigma^N_{i=1}\theta^{i-}_{t+1} \tag{11}$$

$$E[\hat{y}_{t+1}] = \frac{1}{N}\Sigma^N_{i=1}\hat{y}^i_{t+1} \tag{12}$$

Now using the updated parameter, the new model state trajectories (state forecasts) and prediction trajectories are generated:

$$x^{i-}_{t+1} = f(x^{i+}_t, u^i_t, \theta^{i+}_{t+1}) + \omega^i_t \quad \omega^i_t \sim N(0,\Sigma^x_t) \quad \forall \quad Q_t = \Sigma^x_{t+1} \tag{13}$$

$$\hat{y}^i_{t+1} = h(x^{i-}_{t+1}, \theta^{i+}_{t+1}) + v^i_{t+1} \quad v^i_{t+1} \sim N(0,\Sigma^y_{t+1}) \quad \forall \quad R_{t+1} = \Sigma^y_{t+1} \tag{14}$$

Model states ensemble is similarly updated as follows:

$$x^{i+}_{t+1} = x^{i-}_{t+1} + K^x_{t+1}(y^i_{t+1} - \hat{y}^i_{t+1}) \tag{15}$$

$$y^i_{t+1} = y^i_{t+1} + v^i_{t+1} \quad v^i_{t+1} \sim N(0, R_{t+1}) \quad \forall \quad R_{t+1} = \Sigma^y_{t+1} \tag{16}$$

where $K^x_{t+1} \in \mathbb{R}^{m \times n}$ is the Kalman gain for correcting the state trajectories and is obtained by:

$$K^x_{t+1} = \Sigma^{xy}_{t+1}\left[\Sigma^{yy}_{t+1} + R'_{t+1}\right]^{-1} \tag{17}$$

where $\Sigma^{xy}_{t+1} \in \mathbb{R}^{m \times n}$ is the cross-covariance matrix of states ensemble and prediction ensemble (Eq. 16).

$$\Sigma^{xy}_{t+1}(s,j) = \frac{1}{N}\Sigma^N_{i=1}\left[(x^{i-}_{t+1}(s) - E[x^-_{t+1}(s)])(\hat{y}^i_{t+1}(j) - E[\hat{y}_{t+1}(j)])\right] \tag{18}$$

[revised manuscript text omitted]

---

## Author Comment (AC1)

In this study, the authors present a Data Assimilation (DA)-hydrodynamic modeling framework where multiple gauge observations are integrated into the LISFLOOD-FP model to improve the performance of flood inundation mapping. The results indicate that the multivariate assimilation of point-source observations into hydrodynamic models can improve the accuracy and reliability of probabilistic flood inundation mapping by 5-7% while it also provides the basis for sequential updating and real-time flood inundation mapping. This paper is well written, well-organized and has practical meaning for inundation mapping, risk analysis and decision making. I have some minor comments for the authors to improve their paper.

**Response:** We thank the reviewer for the positive evaluation of our work.

Line 168: Xu et al., 2017 is used as a reference. However, this paper is not listed in the bibliography.

**Response:** This reference will be added to the bibliography.

The text (e.g. names of counties and states) in Figure 1 is blurred. Please improve it

**Response:** The quality of figure will be improved in the revised version of the manuscript.

In addition, the water discharge at gauge 1 and 2 as well as water stage at gauge 2 (n=3) are assimilated. The authors could also assimilate the water discharge and the water stage at gauge 1, 2 and 3 to further improve the performance. Is the data available?

**Response:** Thank you for the comment. The water stage at gauge 2 is not available for the peak of Harvey. We used all available data for two internal gauges. In this research we explain the DA-hydrodynamic modeling framework in a generic format where n number of point source observations can be assimilated into a hydrodynamic model. This helps the potential users to follow this approach and apply it for different number of observations in future studies.

Figures 6 & 7: the two figures presented the comparison between OL and EnKF. It would be better to add the real inundation map in the third column of the figures.

**Response:** The reference flood inundation map we use in this study is representative of maximum flood maps during all days (The union of daily flood maps). We don't have daily real inundation maps to compare with simulated maps at daily scale. However, we provided these two figures to explore the differences of OL and EnKF approach for generating probabilistic inundation maps. In order to validate these two methods, we first evaluate the temporal behavior of these two approaches by comparing the simulated time series with observations at gauges (Figure 4) and then test the spatial behavior by taking the union of all simulated daily maps and comparing with the reference map in Figure 8. Although we don't have daily real maps to update Figures 6 and 7, we improved Figure 8 by adding three subplots that display the comparison of maximum OL and ENKF with reference map. Please see the updated figure below:

[Figure]

Line 542-545: Here, we used the EnKF data assimilation method in conjunction with a hydrodynamic model to account for different sources of uncertainties involved in different layers of model simulations, including the boundary conditions, model parameters, and initial condition, and generate real-time probabilistic flood inundation maps. It is nice to consider different sources of uncertainty here. How the uncertainty fluctuates with ensemble size and error settings?

**Response:** Analyzing the sensitivity of DA-hydrodynamic modeling results, to the error settings and ensemble size, is an interesting topic of research. However, to accurately address these two points, a multitude of DA-hydrodynamic models should be simulated within a sensitivity analysis framework which is not within the scope of this study. Here, we aimed to demonstrate the effectiveness of assimilating multi-source data into hydrodynamic models for improving inundation mapping skill. According to several past studies that explored the sensitivity of EnKF to the ensemble size, the model performance is highly sensitive to small ensemble sizes less than 50 but the sensitivity will be reduced by moving to larger numbers around 100 (e.g. see Figure 4 in Moradkhani et al., 2005 and Figure 2 in Gillijns et al., 2006). With access to sufficient number of computing cores at the University of Alabama High Performance Computing cluster, we were able to use a relatively large number of 100 ensemble members (It is larger than most of studies that used EnKF method). This ensures that our results will provide reasonable range of

uncertainties. Regarding the error settings, the synthetic experiment is typically used to tune the perturbation parameters. The synthetic experiment results together with the recommendations of past studies, are the basis for designing the error settings in a DA framework.

Figure 4: it would be better to add the predictive interval of open loop

**Response:** Thanks for the great suggestion. Figure 4 was updated and the open loop interval was added. Please see the updated Figure below:

[Figure]

Line 112: the following paper may be a good reference for the SMAP soil moisture data assimilation. Xu et al. Continental drought monitoring using satellite soil moisture, data assimilation and an integrated drought index. Remote Sensing of Environment, 2020, 250:112028.

**Response:** The reference will be added to the manuscript.

---

## Author Comment (AC2)

The paper is interesting and presents a useful methodology for using data assimilation in flood modeling studies. A few assumptions need to be clarified. Limitations, uncertainties and implications need to be further discussed. The authors should also provide recommendations on future research and how this work can be applied to other study areas. I have recommended several edits and some comments in the PDF. Here are some additional comments:

**Response:** Thank you for the positive and detailed review of our work and for providing constructive comments. Please see our response and revisions below.

Introduction: The section can be shortened and be more concise.

**Response:** We will shorten the introduction and will revise the text according to the suggestion of the reviewer in the pdf file.

Past research on probabilistic flood modeling should be further acknowledged (e.g., Aronica et al. 2012, Savage et al. 2016, Papaioannou et al. 2017, Ahmadisharaf et al. 2018).

**Response:** The suggested references will be added to the revised manuscript.

Please provide more information on the study catchment, particularly those that affect your model results. This includes computational area, soil type, channels' size, ground slope, land use etc.

**Response:** More information regarding the study catchment was added to the revised manuscript. Please refer to lines 197-203. The added text can be seen below.

"The study area is located in the State of Texas (Figure 1.b) in the middle of the San Jacinto watershed (Figure 1.c), a highly developed basin (USGS HUC6 #120401) with the area of 10400 km$^2$. The main channels simulated in the study are around 106 km and drain into three HUC8 watersheds; the Spring (#12040102), West Fork San Jacinto (#12040101) and East Fork San Jacinto (#12040103). The drainage areas of the channels are relatively flat with an average slope of 0.62%, and the soil is mostly impermeable due to the high rate of recent developments in this region."

How well do the 120 m DEM represent the watershed topography and bathymetry? What are potential errors and how they affect your findings?

**Response:** We added a new paragraph in the discussion section explaining the logic behind using a coarse DEM and its potential impacts on the results. Please refer to lines 619-627 and 639-649 in the revised manuscript. The new lines can be seen below.

"For real-time flood inundation mapping, timely decision making is of paramount importance. The time between the issuance of the warning and the occurrence of the flood is typically a short period less than a day. Additionally, the flood waves propagate, inundate the affected regions and cause damages rapidly. Thus, the main requirement for real-time probabilistic inundation mapping is to develop a fast and efficient modeling framework that is beneficial for decision makers and emergency managers. Considering the high computational expense of hydrodynamic models and the need for generating a multitude of simulations in the probabilistic fashion, this

study uses a coarse resolution 120m DEM to maintain the efficiency of the modeling and meet the requirements for practical benefits."

More details on the hydrodynamic model setup is needed. How long was the warm-up period? What was the simulation time step? What was the Courant number?

**Response:** We added more information regarding the model setup. Please refer to lines 356-358 in the revised manuscript. This information can be seen below.

"The simulation period of the LISFLOOD-FP model is set up for 45 days from July-30-2017 to Sep-12-2017 and the entire month of July is used as a warm-up period. The model time step and the Courant number are set to 1 second and 0.7, respectively, and the model is simulated at daily scale."

Please present the computational time of your simulations. In practice and with the existing resources, how practical it is to use the 2D unsteady models for real-time flood forecasting? In particular, high resolution analyses might be desired. Is the presented DA-hydrodynamic modeling framework computationally efficient to be applied in practice?

**Response:** Thank you for the great questions. We addressed these questions by adding more information regarding the computational efficiency of the proposed DA-hydrodynamic modeling framework and its benefits in practice. Please refer to lines 619-635 in the revised manuscript. The added lines can be seen below.

"For real-time flood inundation mapping, timely decision making is of paramount importance. The time between the issuance of the warning and the occurrence of the flood is typically a short period less than a day. Additionally, the flood waves propagate, inundate the affected regions and cause damages rapidly. Thus, the main requirement for real-time probabilistic inundation mapping is to develop a fast and efficient modeling framework that is beneficial for decision makers and emergency managers. Considering the high computational expense of hydrodynamic models and the need for generating a multitude of simulations in the probabilistic fashion, this study uses a coarse resolution 120m DEM to maintain the efficiency of the modeling and meet the requirements for practical benefits. In this study, the DA-hydrodynamic modeling framework is executed on the University of Alabama High Performance Computing (UAHPC) cluster. Considering the ensemble size of 100, we submit a job array with 100 cores where each core is assigned to a specific member of the DA-hydrodynamic modeling simulation. The efficient hydrodynamic model setup with coarse resolution DEM helps to simulate the Harvey and generate probabilistic results in 4-5 hours (~ 4 hours for the hydrodynamic simulation and ~20 minutes for the DA). Applying this computationally efficient framework is highly beneficial, specially for the emergency response agencies (e.g. FEMA), insurance companies, Water Centers, and other private companies that need to forecast the inundation areas and take timely decisions a few hours before the onset of floods."

Please italicize all variables/parameters in the text.

**Response:** Done.

Any rational for considering a normal distribution for the model boundary conditions with a mean of zero and relative error of 20%? If it is derived from previous studies, please point this

out the first place you present this assumption. You should also discuss the validity of your assumption by discussing how close the previous studies are to your study.

**Response:** We added two main references that explain the error of observed flows vary in the range 8-25%. However, the basis for choosing the best error term in the assimilation works is the trial and error and the manual tuning of the values to achieve the most reliable predictions. Please refer to lines 324-327 and lines 368-370 in the revised manuscript.

Any rational for considering a normal distribution for the model initial conditions with a mean of zero and relative error of 1 m? If it is derived from previous studies, please point this out the first place you present this assumption. You should also discuss the validity of your assumption by discussing how close the previous studies are to your study.

**Response:** The error term for the bathymetry is chosen from the expert judgment, trial and error and tuning to achieve the most reliable predictions. It is quite common in DA studies that we choose errors based on manual tuning if there is very little a priori knowledge about the physically correct error terms. We added more information about the logic behind our selections. Please refer to lines 338-340 in the revised manuscript.

Any rational for considering uniform distributions for the channel roughness and bathymetry? Where are the ranges—(0,0.1) and (39,42)—coming from? If it is derived from previous studies, please point this out the first place you present this assumption. You should also discuss the validity of your assumption by discussing how close the previous studies are to your study.

**Response:** For the channel roughness, we already provided the references in the manuscript. The range of bathymetry parameter is again estimated from the expert judgment and trial and error. It is quite common in DA studies that we choose errors based on tuning if there is very little a priori knowledge about the physically correct error terms. Please refer to lines 365-370 in the revised manuscript.

To perturb the uncertainty of parameters via Latin Hypercube Sampling, how many samples were settled? How did you ensure that the samples are sufficient for the numerical convergence?

**Response:** The number of samples is equal to the ensemble size of the experiment. Therefore we generated 100 samples from the uniform distributions. According to several past studies that explored the sensitivity of EnKF to the ensemble size, the model performance is highly sensitive to small ensemble sizes less than 50 but the sensitivity will be reduced by moving to larger numbers around 100 (e.g. see Figure 4 in Moradkhani et al., 2005 and Figure 2 in Gillijns et al., 2006). With access to sufficient number of computing cores in the University of Alabama High Performance Computing cluster, we were able to use a relatively large number of 100 ensemble members (It is higher than most of studies that used the EnKF method). This ensures that our results will provide reasonable range of uncertainties.

The three experiments presented under the Results section, should be clearly defined and described under the Methodology section. As of now, it is hard to follow the difference between the three experiments and details of each.

**Response:** Thanks for the making this point. We revised the manuscript by adding required texts, and changing the headings of both method and results sections.

Figures 6 and 7 should be merged.

**Response:** Using one single figure representing inundation maps for all 6 days reduces the size of subplots. We believe that using smaller subplots for each inundation map will degrade the quality and decrease the usefulness of the figure as the details will not be observable anymore. Thus, we feel it will be better to keep this format as one figure shows the upper limb and the other one highlights the lower limb of the flood hydrograph.

In addition to the inundation extent, it will be useful to compare the performance of EnKF with the OL on other flood characteristics such as depth.

**Response:** We already compared the EnKF and OL performance for simulating water depth and discharge in Figure 4. In fact, our validation strategy includes two components: 1. Validating the dynamics of DA-hydrodynamic modeling framework at point sources (Figure. 4): Here, we focus on water depth and discharge simulated by OL and EnKF and compare their performances. 2. Validating the spatial behavior of DA-hydrodynamic modeling framework (Figure. 8): Here, we compare the flood extents simulated by the OL and EnKF.

Broader impacts need to be discussed. The authors should discuss what implications these results have for urban planners and floodplain managers etc. and what existing programs in the US may benefit from this research.

**Response:** The broader impacts of the proposed DA-hydrodynamic modeling framework for applying in practice was further discussed. Please refer to lines 619-635 in the revised manuscript. The new text can be seen below.

"For real-time flood inundation mapping, timely decision making is of paramount importance. The time between the issuance of the warning and the occurrence of the flood is typically a short period less than a day. Additionally, the flood waves propagate, inundate the affected regions and cause damages rapidly. Thus, the main requirement for real-time probabilistic inundation mapping is to develop a fast and efficient modeling framework that is beneficial for decision makers and emergency managers. Considering the high computational expense of hydrodynamic models and the need for generating a multitude of simulations in the probabilistic fashion, this study uses a coarse resolution 120m DEM to maintain the efficiency of the modeling and meet the requirements for practical benefits. In this study, the DA-hydrodynamic modeling framework is executed on the University of Alabama High Performance Computing (UAHPC) cluster. Considering the ensemble size of 100, we submit a job array with 100 cores where each core is assigned to a specific member of the DA-hydrodynamic modeling simulation. The efficient hydrodynamic model setup with coarse resolution DEM helps to simulate the Harvey and generate probabilistic results in 4-5 hours (~ 4 hours for the hydrodynamic simulation and ~20 minutes for the DA). Applying this computationally efficient framework is highly beneficial, specially for the emergency response agencies (e.g. FEMA), insurance companies, Water Centers, and other private companies that need to forecast the inundation areas and take timely decisions a few hours before the onset of floods."

Study limitations and potential areas for future research need to be expanded.

**Response:** The final paragraph of the manuscript was expanded by adding limitations while suggesting more areas of research for future studies. Please refer to lines 667-672 in the revised manuscript. The new paragraph can be seen below.

"The time dependency that exists between the upstream and downstream gauges along a channel can affect the performance of multivariate assimilation with those gauges. For future studies, using a more advanced DA technique that fully characterizes the model structural uncertainty (Abbaszadeh et al., 2019), and considering the time lag dependency between multiple gauges can improve the performance of modeling and provide more realistic assimilation of the hydrodynamic models. Another limitation of this study is the simple assumptions made for perturbing the initial condition (water depth), parameters (channel roughness and river bathymetry) and observations (WSE and discharge). More investigation on the physically meaningful distribution of these values can enhance the performance of the DA-hydrodynamic modeling framework in future studies. A joint assimilation of point source gauges and remotely sensed data can also improve the reliability and accuracy of the results. Finally, proposing a DA-hydrodynamic modeling framework that considers the DEM and channel width uncertainty can provide a more comprehensive uncertainty quantification for probabilistic flood inundation mapping in future studies."

Please discuss how your presented DA-hydrodynamic modeling framework can be used in other study areas. What considerations should be taken to do so?

**Response:** Thanks for the great suggestion. We added a new paragraph to the end of the manuscript where the required considerations for applying the proposed framework to other studies are further discussed. Please refer to lines 675-686 in the revised manuscript. The added paragraph can be seen below.

"An advantage of the proposed DA-hydrodynamic modeling framework is its generic format so that other studies can follow the flowchart in Figure. 2 and use information in Section 3.2 and 3.3 to set up the hydrodynamic model and the EnKF algorithm, respectively. To properly apply this framework to other studies, first, the point source observations of WSE and discharge should be available at daily/sub-daily scales. In other words, the proposed framework cannot be implemented in ungauged basins.  Second, the modeler should have access to high performance computing facilities for parallel simulation of ensemble members. Third, the hydrodynamic model should be sequentially executed within the DA algorithm. The modeler should check the hydrodynamic model manual and make sure that the outputs and initial conditions can be upgraded in a sequential manner. Taking these three considerations into account, the proposed DA-hydrodynamic modeling framework can be applied to any other study areas that are prone to frequent flooding and provide a robust and generic tool for real-time probabilistic flood inundation mapping".

**References:**

Gillijns, S., Mendoza, O.B., Chandrasekar, J., De Moor, B.L.R., Bernstein, D.S., Ridley, A., 2006. What is the ensemble Kalman filter and how well does it work?, in: 2006 American Control Conference. Presented at the 2006 American Control Conference, p. 6 pp.-. https://doi.org/10.1109/ACC.2006.1657419

Moradkhani, H., Sorooshian, S., Gupta, H.V., Houser, P.R., 2005b. Dual state–parameter estimation of hydrological models using ensemble Kalman filter. Adv. Water Resour. 28, 135–147. https://doi.org/10.1016/j.advwatres.2004.09.002

---

## Author Response (AR2)

We would like to thank the editor for the final check and finding minor errors in our manuscript. We corrected equations 20-23 and addressed other minor comments. Regarding the comments of the editor on using sigma sign for covariance, we need to emphasize that this is not a summation, instead it is just a variable shown with this symbol. In the literature of Data Assimilation, the sigma sign is used to define the covariance. We use the same terminology to be consistent with the literature. Here we refer the editor to three pioneer papers in the DA that used the same symbol for defining the covariance (Anderson, 2001; Daescu and Langland, 2013; Moradkhani et al., 2005).

Anderson, J.L., 2001. An Ensemble Adjustment Kalman Filter for Data Assimilation. Mon. Weather Rev. 129, 2884–2903. https://doi.org/10.1175/1520-0493(2001)129<2884:AEAKFF>2.0.CO;2

Daescu, D.N., Langland, R.H., 2013. Error covariance sensitivity and impact estimation with adjoint 4D-Var: theoretical aspects and first applications to NAVDAS-AR. Q. J. R. Meteorol. Soc. 139, 226–241. https://doi.org/10.1002/qj.1943

Moradkhani, H., Sorooshian, S., Gupta, H.V., Houser, P.R., 2005. Dual state–parameter estimation of hydrological models using ensemble Kalman filter. Adv. Water Resour. 28, 135–147. https://doi.org/10.1016/j.advwatres.2004.09.002